# Real-space observation of the dissociation of a transition metal complex and its concurrent energy redistribution

Aviad Schori[1], Elisa Biasin [1,2], Ambar Banerjee[3,4], Sébastien Boutet [5], Philip H. Bucksbaum[1,6], Sergio Carbajo[5], Kelly J. Gaffney [1,7], James M. Glownia[5], Robert Hartsock[1], Kathryn Ledbetter[1], Andreas Kaldun[1], Jason E. Koglin[5], Kristjan Kunnus[1], Thomas J. Lane [5], Mengning Liang [5], Michael P. Minitti [5], Jordan T. O'Neal [1], Robert M. Parrish[1], Frédéric Poitevin[1], Jennifer M. Ruddock[8], Silke Nelson[5], Brian Stankus[8,9], Peter M. Weber [8], Thomas J. A. Wolf [1], Michael Odelius [3] & Adi Natan [1] ✉

Mechanistic insights into photodissociation dynamics of transition metal carbonyls, like $Fe(CO)_5$, are fundamental for understanding active catalytic intermediates. Although extensively studied, the structural dynamics of these systems remain elusive. Using ultrafast X-ray scattering, we uncover the photochemistry of $Fe(CO)_5$ in real space and time, observing synchronous oscillations in atomic pair distances, followed by a prompt rotating CO release preferentially in the axial direction. This behavior aligns with simulations, reflecting the interplay between the axial Fe-C distances' potential energy landscape and non-adiabatic transitions between metal-to-ligand charge-transfer states. Additionally, we characterize a secondary delayed CO release associated with a reduction of Fe-C steady state distances and structural dynamics of the formed $Fe(CO)_4$. Our results quantify energy redistribution across vibration, rotation, and translation degrees of freedom, offering a microscopic view of complex structural dynamics, enhancing our grasp on $Fe(CO)_5$ photodissociation, and advancing our understanding of transition metal catalytic systems.

The use of light to drive synthesis has seen tremendous growth in the last decade[1,2]. In particular, organometallic photocatalysts such as transition metal carbonyls[3–5], are pivotal for understanding the mechanisms of metal-ligand bond breakage upon light exposure, a cornerstone issue in chemistry with implications in synthesis and catalysis. The efficiency and selectivity of photocatalytic processes are directly related to the molecular-level mechanisms of energy flow and transformation, including internal vibrational relaxation, rotational motions, and dissociation events in these systems[6,7]. By understanding these ultrafast structural dynamics, chemists can design more effective catalytic systems and optimize reaction conditions for practical applications.

[1]Stanford PULSE Institute, SLAC National Accelerator Laboratory, Menlo Park, CA, USA. [2]Physical Sciences Division, Pacific Northwest National Laboratory, Richland, WA, USA. [3]Department of Physics, Stockholm University, Stockholm, Sweden. [4]Research Institute for Sustainable Energy (RISE), TCG Centres for Research and Education in Science and Technology, Kolkata, India. [5]Linac Coherent Light Source, SLAC National Accelerator Laboratory, Menlo Park, CA, USA. [6]Department of Physics, Stanford University, Stanford, CA, USA. [7]Department of Chemistry, Stanford University, Stanford, CA, USA. [8]Department of Chemistry, Brown University, Providence, RI, USA. [9]Department of Chemistry and Biochemistry, Western Connecticut State University, Danbury, CT, USA. ✉e-mail: natan@stanford.edu

**Fig. 1 | Photodissociation dynamics and structural changes in Fe(CO)$_5$ as revealed by pair density analysis. a** Schematic depicting Fe(CO)$_5$ photochemistry: UV (266 nm) pulse excites Fe(CO)$_5$ from its ground state (GS) to a metal-to-ligand charge-transfer excited state (MLCT). This transition leads to periodic crossovers to the dissociative metal-centered (MC) state, observed through a symmetric stretch mode in axial Fe-C bonds. Each MLCT to MC crossover may induce dissociation of the first CO ligand (inset, dashed arrows represent crossover), occurring approximately every 100 fs, generating hot Fe(CO)$_4$ and a rotating dissociating CO. Subsequently, a second CO is dissociating at a 3 ps timescale. **b** Structural dynamics are revealed through atom pair density (PD) analysis: The different atom-pair types (color-coded on Fe(CO)$_5$ structure) are weighted by their charge products. We compare the steady-state PD at $\tau = 0$ (solid black) with the PD at 250 fs after photoexcitation (dashed line). The latter shows a shift in density toward longer distances and a corresponding reduction at the steady-state peaks. Subtracting the steady-state PD from the 250 fs distribution yields the difference pair density $\Delta PD$ (orange), which is used to trace the structural changes over time.

Extensive research on iron pentacarbonyl Fe(CO)$_5$, particularly on processes triggered by metal-to-ligand charge-transfer (MLCT), has used various methods to explore its photochemistry revealing the dissociation of carbon monoxide (CO) from Fe(CO)$_5$ into Fe(CO)$_4$ and then Fe(CO)$_3$ in gas phase dynamics[8–14]. The initial experimental studies used 266 nm UV pulses to photoexcite Fe(CO)$_5$ in the gas phase and probed its dynamics using optical ionization[9–11] measuring various time constants up to the few ps range, without assigning the different possible species. However, a sequential singlet pathway was suggested, with the assumption that intersystem crossings are not viable in such timescales. The suggested pathway was prompt dissociation from excited singlet-state Fe(CO)$_5$ to the lowest energy singlet state of Fe(CO)$_4$ ($^1A_1$), following which, a second CO dissociation to Fe(CO)$_3$ with a time constant of 3 ps. Time-resolved electron diffraction experiments[15] validated the singlet-state Fe(CO)$_4$ using two-photon absorption at 620 nm and a temporal resolution of 10-20 ps. Direct evidence of the singlet and sequential aspect of the dissociation of multiple CO ligands has been established with a temporal resolution of 1 ps[12] using time-resolved valence and core-level photoelectron XUV spectroscopy.

Recently, a theoretical study suggested that the MLCT transition causes Fe-C bond oscillations in the trigonal bipyramidal complex[16]. As a result, the first CO dissociation process involves a probabilistic sequential dissociation, that occurs each time non-adiabatic transitions happen between manifolds of bound MLCT and dissociative metal-centered (MC) excited states (Fig. 1a).

While studies probed the population transfer between electronic states, to fully understand how photon energy transforms into both electronic and nuclear dynamics, it's crucial to observe ultrafast structural dynamics, which requires probing atomic motions on the scale of angstroms and femtoseconds. Such direct observation is key to understanding the creation of specific reactive intermediates, and despite its importance, achieving this level of detail has so far proved difficult.

Ultrafast X-ray scattering (UXS) and MeV ultrafast electron diffraction (UED) studies have become viable tools to uncover ultrafast

structural dynamics of increasing complexity, from wavepacket vibration and dissociation of diatomic molecules to chemical reactions and structural changes of molecules in solution environments[17–31]. Electron diffraction offers a greater momentum transfer range ($Q_{max} \sim 12$ Å$^{-1}$) that allows in principle the direct inversion to real space of the scattering signal, but often at lower temporal resolution. The real-space resolution, $\Delta R \sim \pi/Q_{max}$, is determined by the maximum Q-range probed. The more restricted $Q_{max}$, the coarser the resolution in R-space, potentially leading to inversion artifacts. Ultrafast X-ray pulses from free electron lasers (FELs) often access better temporal resolution but with a restricted range ($Q_{max} \sim 4$ Å$^{-1}$) that has limited the inversion and interpretation of the scattering signals and required complementary support through modeling and simulations[32–34]. Recent improvements in inversion methods that account for limitations originating from the measurement, including a restricted Q-range[35], enable the robust real-space inversion central to this study. Briefly, the improved inversion method views scattering from the perspective of real-space sampling, where each sampling point corresponds to a scattering kernel that describes how that sampling distance translates into the measured scattering pattern and its inversion under deterministic experimental constraints (Supplementary Note 4). As a result, this method can better reconstruct the underlying structural changes even when the available momentum transfer window is restricted, a key advantage for ultrafast studies at FELs, where achieving both high temporal resolution and a wide Q-range remains challenging.

Here, we demonstrate the use of UXS and real-space inversion to directly observe the photochemistry of Fe(CO)$_5$ with atomic resolution in space and time. We observe its initial coherent motion, prompt dissociation that includes molecular rotation, and the rovibration dynamics of the intermediate Fe(CO)$_4$. We also observe and quantify the secondary dissociation to Fe(CO)$_3$. We resolve multiple simultaneous atomic motions on femtosecond timescales of a first-row transition metal complex in the gas phase, where charge density distance distributions from different atom pairs contribute similarly to the signal (Fig. 1b), making it the first observation, to our knowledge, of

ultrafast gas phase scattering from a transition metal complex. We observe microscopic real-space details of coherent to thermal pair density dynamics, which is an intuitive and independent probe for processes that take place across timescales, despite the restricted momentum transfer range measured ($Q_{max} = 4.34$ Å$^{-1}$). The time-resolved real-space information is recovered in a model-free way, which enables a straightforward comparison with ab initio simulations, offering a path for robust validation of theory for the challenging case of transition-metal excited-state dynamics.

## Results and Discussion

For non-periodic samples such as in gas or the solution phase, assuming the independent atom model, the scattering information of typical experiments only encodes the distances between each pair of charges captured by the pair density function. The pair density of Fe(CO)$_5$ includes five distinct types of atom pairs spanning intramolecular distances of 1 to 6 Å, as shown in Fig. 1b. To capture the evolving structural dynamics, we analyze the atomic pair charge density difference $\Delta PD(R, \tau) = PD(R, \tau)_{on} - PD(R)_{off}$, which is obtained by subtracting the laser-off reference signal from the optical pump laser-on signal at a delay time $\tau$ following photoexcitation.

In Fig. 2a, we present the early excited-state dynamics of Fe(CO)$_5$ as captured by $\Delta PD$. This simulation is based on semi-classical excited-state molecular dynamics and is limited to the initial CO loss process, covering delays up to 640 fs (Supplementary Note 2). Positive intensities indicate new distances with a magnitude corresponding to the charge product of the contributing atom pairs, and negative peaks indicate the drop in steady-state pair distances. As a result, even though the motion originates in the Fe-C bond distances (region $\alpha$), a change in Fe-C pair distances affects many other atom pair distances that act as "spectators", as seen during the first 200 fs (region $\beta$). This is because atoms of type X that remain stationary to the Fe-C motion, will contribute to correlated Fe-X and C-X motions, effectively amplifying the original Fe-C motion. We also note that the contribution to $\Delta PD$ is distributed across several types of atomic pairs that can also overlap. For example, a distinct $\Delta PD$ is found around 4 Å as a result of multiple C-O and O-O pairs that are spectators to the early dynamics. Early on, the CO dissociation overlaps with other pair distances and seems to be obscured by them. However, it becomes noticeable at larger pair distances in the region $\gamma$, where the dissociated CO leads to pair distances that exceed the intramolecular distances of the remaining Fe(CO)$_4$.

We use the simulated pair density dynamics to obtain the simulated scattering signal (Supplementary Note 3, Supplementary Fig. 1) and convolve it with the estimated finite instrument response (58 fs, Supplementary Note 8, Supplementary Fig. 13). It is then compared to the experimental scattering results in Fig. 2b. We find that there is reasonable agreement with some minor observed differences that can be attributed to the method used to subtract the experimental background and the limitation of the simulation to capture later dynamics that include the second dissociation (Supplementary Notes 1,2). Notably, we observe a time-dependent evolution of the scattering difference signal at Q < 0.9 Å$^{-1}$, where a negative peak forms and shifts toward lower Q values. This suggests charge-density changes on a molecular size scale $2\pi/Q > 7$ Å, consistent with ongoing dissociation. Meanwhile, a positive peak emerges in the 1.3 < Q < 1.9 Å$^{-1}$ range, hinting at structural rearrangement and relaxation over 3.3 Å – 4.6 Å. Although molecular dynamics are often discussed in terms of localized contributions in Q-space, we demonstrate that only by inverting the scattering data into real space, an agnostic linear operation, can we obtain a clearer, resolution-correct interpretation of the underlying dynamics.

We take the simulated scattering signal and then enforce on it the same constraints as we have for the experiment, and transform it to real space using the method in[35] to obtain the pair density dynamics given the available Q-range (Supplementary Note 4). In Fig. 2c, we

show the impact of these constraints on the extracted structural dynamics information, highlighting how the spatial resolution of the simulated $\Delta PD$ is limited by the experimental Q-range. We observe, that while individual pair distances are not resolved, the underlying dynamics are still recovered. The dominant pair density difference is enhanced by the spectator C-O and O-O pairs at distances of 4 < R < 6 Å ($\beta$ region) that capture the onset of synchronous oscillations that originate in the Fe-C pairs. The dissociation is first manifested as an increase of density at these distances at a 200-400 fs delay, following positive contributions to distances > 7 Å ($\gamma$ region) at later delays. In addition, we observe a modulation in $\Delta PD$ as it evolves from region $\beta$ to $\gamma$. We will discuss these effects and their relation to the rotation of the dissociated CO.

Figure 2d presents the experimental $\Delta PD$ obtained from the inversion of the measured scattering differences. In the 50-200 fs window, both experiment and simulation show matching pre-dissociation oscillations at comparable pair distances and frequency. From 200–400 fs ($\beta$ region), we observe a clear enhancement in $\Delta PD$ that aligns with the simulated CO-dissociation pathway, and in the 450-600 fs range ($\gamma$ region), the measured $\Delta PD$ modulations agree with the theoretical pair-distance signals. Overall, the positive and negative measured density features follow the patterns predicted by the simulation, highlighting the consistency between experiment and theory over the simulated pump-probe time frame.

Moreover, the experimental $\Delta PD$ extends beyond the simulation's accessible timescales, providing insights into longer-term structural dynamics. A main characteristic that is noticeable at longer delays ($\tau > 400$ fs) is the further depletion of pair density around the $\alpha$ region that coincides with the steady-state distance of the Fe-C pairs. The ongoing reduction in pair density within this region can be only attributed to a second CO dissociation and the dynamics of Fe(CO)$_4$ that was formed. Additionally, the rate at which density decreases in the region $\alpha$ corresponds to an increase in density between 6.5 < R < 8 Å for times greater than 700 fs.

When analyzing the measured scattering anisotropy difference for a lower Q-range (averaged for Q < 0.9 Å$^{-1}$) (Fig. 2b inset), we find that the anisotropy sign and temporal behavior favors the axial direction for the first CO loss (Supplementary Note 3, Supplementary Fig. 2), following an anisotropy sign flip at longer delays, indicating that subsequent Fe(CO)$_4$ dynamics and the second dissociation is affected more by equatorial than axial CO ligands (Supplementary Note 1). This agrees with the calculation in ref. [16] that predicts that the MLCT transition initiates motion with a preferential loss of axial CO (Supplementary Note 3).

In Fig. 3, we average the experimental $\Delta PD$ across distance regions ($\alpha$, $\beta$, $\gamma$) and compare them to the simulated case, up to their termination time (640 fs). The range in region $\alpha$ provides an opportunity to selectively probe the Fe-C dynamics, being the only pair type in that range. To account for total CO production via dissociation, we consider the averaged $-\Delta PD$ in the region $\alpha$, as every CO leaving the molecule further reduces the Fe-C density of the steady-state distances.

The temporal evolution of the experimental $\Delta PD$ is then used to fit a kinetic rate model that is based on a previous study[12] and revised to account for the time delay in the onset of dissociation, as detailed in Supplementary Note 5. For the first CO dissociation, we obtain a dissociation onset delay of $\tau_0 = 43 \pm 4$ fs due to predissociation oscillatory motion, and a dissociation time constant of $\tau_1 = 101 \pm 13$ fs. For the second dissociation, we obtain a time constant of $\tau_2 = 3 \pm 0.5$ ps, consistent with recent studies[12,13]. While the predissociation oscillations are less visible in that range, their main signature is better observed in pair distance region $\beta$ (3.7 < R < 7 Å). In this distance range, the first CO loss is marked by a density peak at 250 fs, coinciding with the Fe-C and Fe-O pairs crossing this range. The simulated $\Delta PD$ agrees well with the measured density in all ranges up to the point the second

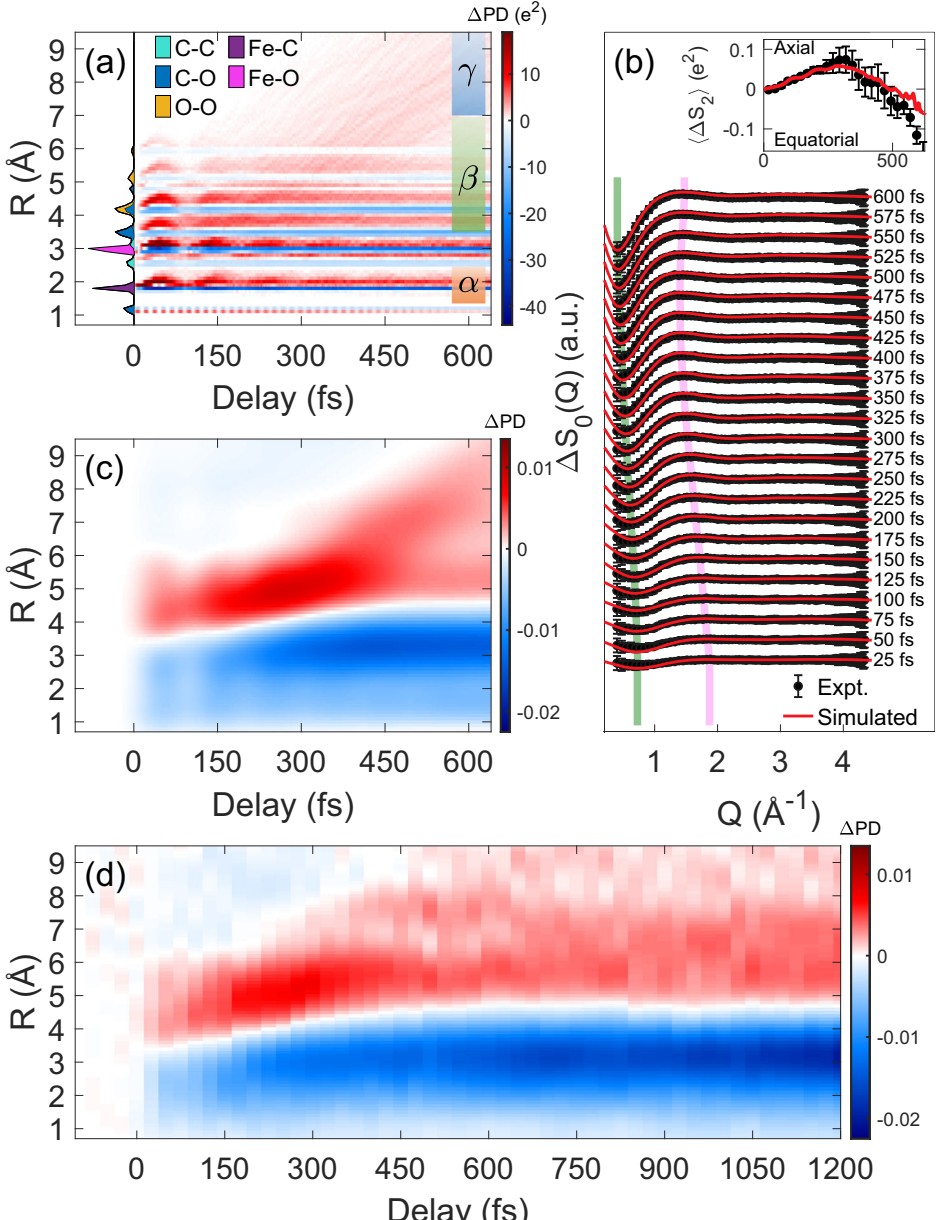

**Fig. 2 | Comparison of simulated and experimental pair-density and scattering data reveals dissociation and structural rearrangements. a** The simulated pair density difference $\Delta PD$ for photoexcited $Fe(CO)_5$ obtained from averaging 103 trajectories (Supplementary Note 2), and derived by subtracting the steady state pair density (perpendicular area plot, colors encode atom pair types) from the time-dependent density. The motion initiates within the Fe-C bond ($\alpha$ region), however, it is also observed across most pairs due to the spectator effect (see text). **b** Simulated and experimental isotropic scattering difference signal $\Delta S_0$ for photoexcited $Fe(CO)_5$ in momentum-transfer space for a series of time delays. Error bars derive from each time bin's weighted sample variance via a weighted least-squares fit, as detailed in[19]. Dissociation is marked by amplitude shift to lower $Q$ and depletion of the negative peak (green stripe), while structural rearrangement forms a positive peak evolving at $1.3 < Q < 1.9$ Å$^{-1}$ (purple stripe). (inset) The averaged anisotropy of the scattering signals $\langle \Delta S_2 \rangle$ for $Q < 0.9$ Å$^{-1}$ (error bars represent the standard deviation of this average) shows that axial CO is more likely to be first lost. (Supplementary Note 3). **c** The inversion of the simulated scattering given the detectable $Q$ range, reproduces the PD oscillations in the $\beta$ region (3.7-7 Å) following a 200-400 fs density increase, consistent with the anisotropy dynamics. At longer delays, the dissociating PD separates from the parent molecule in the $\gamma$ region ($R > 7$ Å), where the ~ 7.5 Å modulation is attributed to CO rotation. **d** Experimental real-space inversion confirms these early dynamics and extends to longer times, capturing a second CO loss for delays > 700 fs, marked by further density depletion (increase) in the $\alpha$ ($\gamma$) region.

CO loss becomes visible at ~ 500 fs. For region $\gamma$ ($7 < R < 10$ Å), we track the time required for the dissociating pair density to exceed the intramolecular distances of $Fe(CO)_5$. The onset of the second dissociation is marked by a slower density increase beginning at a delay of $\tau > 950$ fs. This density growth, with a time constant of 3 ps, is consistent with that observed in region $\alpha$.

When the experimental $\Delta PD$ is subtracted from the kinetic model fit in distance range $\alpha$, the residual $\delta(\tau)$ reveals distinct oscillations in the 400-1200 fs range (Fig. 3d), with a prominent frequency of $78 \pm 16$ cm$^{-1}$. We attribute this motion to C-Fe-C bending modes of the hot $Fe(CO)_4$. This interpretation is supported by a normal mode analysis of the $Fe(CO)_4$ molecule considering various ground state configurations (singlet open shell, singlet closed shell, and triplet). Each normal mode's motion was translated to its corresponding pair density dynamics using the experimental constraints to determine their relative contributions to the observed frequencies in Fig. 3d. We find that

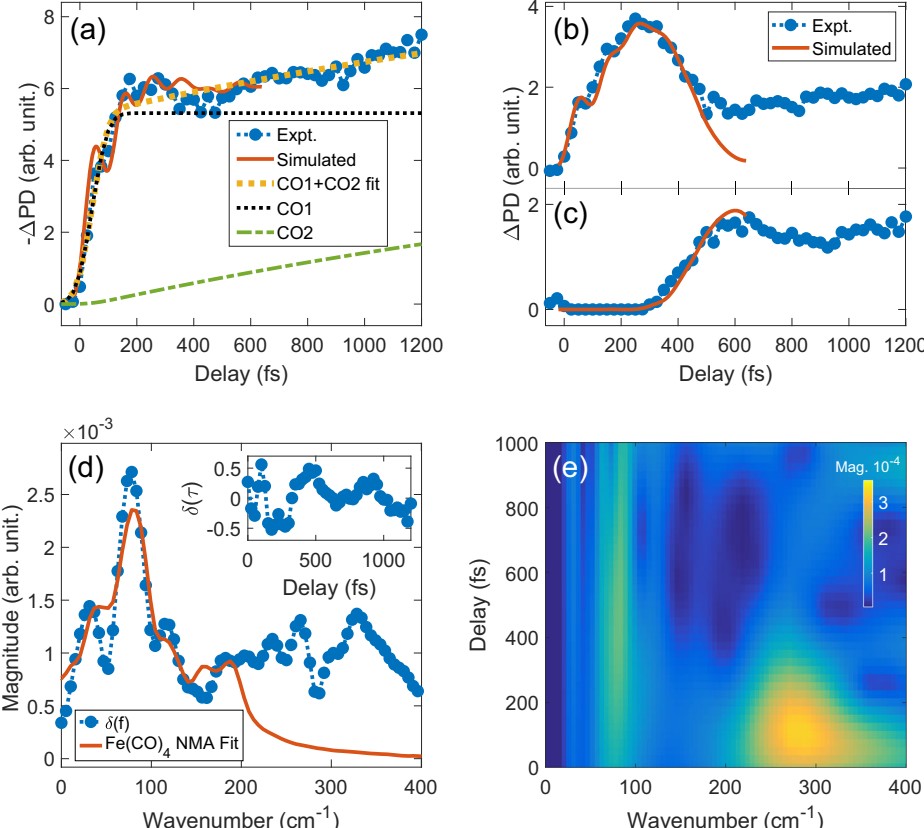

**Fig. 3 | Dissociation of CO ligands and the resulting Fe(CO)$_4$ dynamics across multiple pair distances. a** Within the range $1.3 < R < 2.3$ Å, steady-state Fe-C pair density tracks CO losses and ground-state Fe-C motions. A kinetic model (dashed yellow) fitted to the measured $-\Delta PD$ averaged in this range (blue), and compared to simulated pair-density dynamics (red), which only account for the first CO loss. The model yields rates of $101 \pm 13$ fs (dotted black) and $3 \pm 0.5$ ps (green) for the first and second dissociation, respectively (see text). **b** Predissociation oscillations are observed at $3.7 < R < 7$ Å, with the first CO loss marked by a strong peak around 250 fs as Fe-C and Fe-O pairs traverse this distance range. **c** For $R > 7$ Å, the first CO loss occurs after 450 fs, followed by the second CO loss visible after 1 ps. **d** Fourier analysis of $\delta(\tau)$, the residual between the fitted CO1+CO2 and measured PD in $\alpha$ (inset) reveals a peak at $78 \pm 16$ cm$^{-1}$, attributed to C-Fe-C bending modes in hot Fe(CO)$_4$, as determined by normal mode analysis fit (Supplementary Note 7). **e** Wavelet analysis of $\delta(\tau)$ shows the delayed onset of the 78 cm$^{-1}$ component relative to the prompt 266 cm$^{-1}$ due to predissociation oscillations.

the low-frequency structure (<100 cm$^{-1}$) is predominantly (57%) due to singlet closed shell modes at 42 and 79.5 cm$^{-1}$, with contributions from (29%) singlet open-shell (77.7 and 91.1 cm$^{-1}$), and (14%) triplet (79.2 and 83.9 cm$^{-1}$) modes (Supplementary Note 7, Supplementary Fig. 12). Our findings are consistent with a recent study that shows that both singlet and triplet states are involved in the photodissociation of gas-phase Fe(CO)$_5$[14].

We further support our findings by referencing earlier studies[36,37] which identified Fe(CO)$_5$ Raman allowed bending modes at 74.3 and 97.3 cm$^{-1}$. Contrary to the immediate onset expected from a Raman transition, our observations reveal a delay larger than $\tau_0 + \tau_1 \approx 140$ fs for the measured density oscillations that correspond to the main frequency peak at 78 cm$^{-1}$ (inset of Fig. 3d). This delay, which deviates from the instantaneous response expected of a Raman transition induced by single-pulse photoexcitation, suggests that it originates from the formation of Fe(CO)$_4$. To elucidate these temporal features, in Fig. 3e we present a continuous wavelet transform of $\delta(\tau)$, which shows the prompt onset of the 266 cm$^{-1}$ frequency component that captures the predissociation oscillations and the delayed rise of the 78 cm$^{-1}$ mode. From the wavelet analysis, we can also estimate the relative phase shift between these two frequency components at the average dissociation onset time to be $1.18 \pm 0.22$ rad, corroborating our conclusions.

In the experimental $\Delta PD$ in Fig. 2d we observe a density modulation between region $\beta$ and $\gamma$ at 300-600 fs, which is also captured in the simulation. This modulation is attributed to the CO rotation as it

dissociates, which involves the alternating crossing and separation of dissociative pairs, each consisting of an atom from the remaining Fe(CO)$_4$ and one from the dissociating CO, weighted by their charge product, where the Fe-X pairs dominate the signal.

To illustrate the impact of rotation on $\Delta PD$, we examine the Fe-X pair distances from a single trajectory simulation, as shown in Fig. 4a. Before dissociation, the Fe-C distance is shorter compared to the Fe-O distance. During dissociation, As the CO rotates, the distances equalize after a quarter rotation, and with another quarter turn, the Fe-C distance exceeds the Fe-O distance. In Fig. 4b, we show how these alternating crossing and separation of pair distances appear as density modulations for the simulated $\Delta PD$ using the experimental conditions. In Fig. 4c, we compare the measured $\Delta PD$ modulations with the simulated positions and times of dissociative pair distance crossings. These are weighted by their charge density product and averaged across all trajectories. We observe agreement between the positions and times of the density modulation and the pair distance crossings. In addition, we observe that the initial enhancement in density in the range $\beta$ is not only due to the Fe-C pair crossing other intramolecular pairs distances but also due to the CO rotation motion. The enhancements in $\Delta PD$ within regions $\beta$ and $\gamma$, confirmed by simulation, help us estimate the dissociation velocity and rotational frequency of the molecules (Fig. 4d), as detailed below.

The dissociation velocity and rotational frequency of CO are quantified and compared in Fig. 4d using both simulation and experimental data. From the simulations, we calculate the dissociation

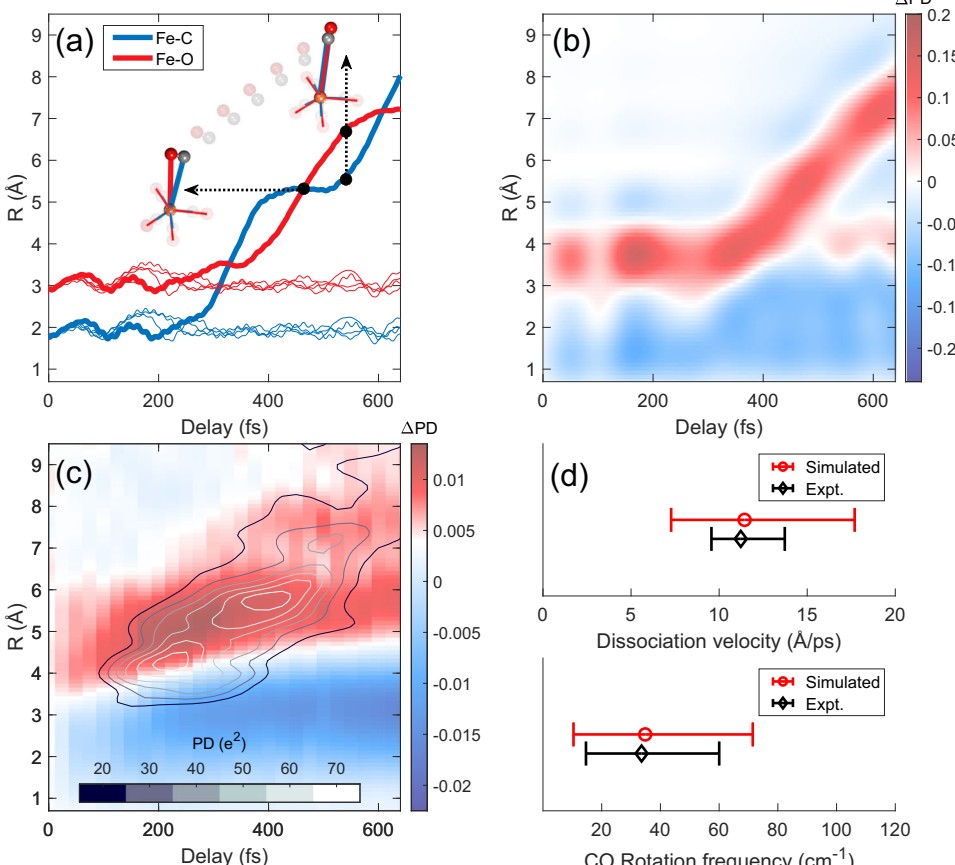

**Fig. 4 | The effect of rotational dissociation on pair dynamics. a** A single trajectory calculation illustrates how CO dissociation and rotation affects the pair distances from a reference atom Fe to the C and O atoms. Fe-C (blue line) and Fe-O (red line) distances cross (horizontal arrow) and separate (vertical arrow) in space and time as the dissociating CO rotates 90°, which in turn produces the density modulations simulated in **b** using the experimental conditions. **c** We compare the average simulated pair density, evaluated at the spatiotemporal intersections of pair distances across all trajectories (color scale refers to contour lines), to the measured $\Delta$PD (red-blue), correlating the observed density modulations with rotational dissociation. **d** The dissociation velocity and rotation frequency are estimated from the measured $\Delta$PD modulations and compared to the expectation values from the simulation averaging all trajectories. Confidence intervals for the simulated data (red) are the widths of the dissociation velocity and rotation frequency distributions derived from all trajectories, while those for the experimental data (black) are obtained by weighting slopes inferred from the kinetic-model-predicted dissociation onset and experimental density modulations (Supplementary Note 6).

velocity by averaging the velocities between the dissociating CO center of mass and the remaining Fe(CO)$_4$ center of mass across all trajectories. Additionally, we determine the rotational frequency of CO by averaging across all trajectories. The analysis shows a correlation between dissociation velocity and rotation frequency, detailing how energy is divided between translational and rotational motion. We obtain an average dissociation velocity of 11.4 Å/$ps$ and rotational frequency to be 35 cm$^{-1}$ (Supplementary Note 6, Supplementary Fig. 7).

Experimentally, the dissociation velocity is estimated by analyzing the onset and progression of dissociation for the Fe-C pair (50-150 fs at 1.8 Å), correlating with the density modulation observed at 400-600 fs and 7-8 Å. The dissociation velocity for CO is thus estimated to be $11.2^{+1.7}_{-2.5}$ Å/$ps$. The CO rotational frequency is inferred to be $34^{+26}_{-19}$ cm$^{-1}$, supported by observed modulations that correspond to contributions of 90° and 270° rotation (Supplementary Note 6, Supplementary Fig. 8). These results agree with the expectation values that are obtained from the trajectory simulations, and we obtain that the kinetic energy release is partitioned at a 10:1 ratio translational (182 meV) to rotational (17.5 meV). Furthermore, we can assign an upper bound for the dissociation velocity of the second CO loss, using an extended time window post-initial CO loss based on the kinetic model (150-250 fs) at the Fe-C steady state distance, and the appearance of the $\Delta$PD enhancement for 6.5 < R < 8 at 700-800 fs. We estimate a

dissociation velocity upper bound of $8.5^{+1.2}_{-0.8}$ Å/$ps$, and a kinetic energy release of 107 meV (Supplementary Fig. 9). The partitioning of energy into rotational and translational degrees of freedom arises from the interplay of the potential energy surface and the coupling of stretch and bend modes. In particular, the pronounced rotational motion of the dissociating ligand directly stems from the coupling of the initial stretching mode that is excited to the broader bend modes of the molecule, effectively funneling vibrational energy into a dissociative rotation pathway as one of the CO ligands departs.

In conclusion, using UXS and real-space inversion we recover the ultrafast changes in the atomic distances of photoexcited Fe(CO)$_5$. We observed the photoinduced structural oscillations that lead to photodissociation, using the spectator effect, a new approach for observation of atomic motion, captured by the correlated pair density dynamics. This effect can enable tracing motion in complex settings using a deterministic local oscillator to benchmark motion for other pair distances. Moreover, we identify the signature of rotation on the pair density dynamics, recover a correlated rotational motion on the first dissociation process, and characterize its rate and energy release distribution, as well as details of the second thermal dissociation process. A core challenge to understanding chemical reaction mechanisms is determining the nature and duration of deterministic and concerted reaction dynamics prior to the onset of statistical and

thermal dynamics. Addressing this challenge requires understanding the dynamics of energy redistribution between the electronic and nuclear degrees of freedom (vibrations, rotations, and translations). By directly accessing all structural degrees of freedom, ultrafast X-ray scattering inverted into real-space and combined with molecular dynamics simulations brings us closer to addressing the challenge of understanding the non-equilibrium flow of energy during chemical reactions.

## Methods

### Ultrafast X-ray scattering

We performed time-resolved gas-phase X-ray scattering experiments at the Coherent X-ray Imaging instrument[38] of the Linac Coherent Light Source (LCLS) at the SLAC National Accelerator Laboratory as previously described[39,40]. The details regarding Data collection and initial scattering analysis are described in Supplementary Note 1. Briefly, $Fe(CO)_5$ (Sigma-Aldrich) was introduced as a room-temperature gas with a pressure of 3 torr and excited by an ultrashort 266-nm pump pulse, and probed by an ultrashort 9.5 keV X-ray pulse. A Cornell-SLAC pixel array detector[41] recorded single-shot scattering patterns, which were binned by their time relative to the pump laser and averaged over multiple shots. The patterns were sorted using a jitter correction timing tool and averaged to form pump-probe delay bins with similar SNR to an effective 25 fs resolution. The raw detector signal was corrected for the scattering geometry and X-ray polarization, and calibrated using a static scattering signal from sulfur hexafluoride ($SF_6$)[17]. The time-delayed scattering signal is subtracted from the signal of the unexcited sample, to allow tracing changes in signal positions and cancel background signals $\Delta S(Q, t) = S_{on}(Q, t) - S_{off}(Q)$. To account for a slowly increasing Fe fluorescence background, each laser-on shot was matched to an averaged laser-off shot under similar X-ray pulse conditions, considering the lab time, electron-beam energy, total scattering intensity, and X-ray diode current. We performed pump pulse energy scans to confirm that we were in the linear absorption regime, and based on the relative signal levels we estimated an excitation fraction of 8% and an instrument response function of 58 fs (Supplementary Notes 8, 9, Supplementary Fig. 13). To increase the fidelity of the scattering signal we used its cylindrical symmetry property by four-folding and applying a weighted averaging of the scattering image quadrants. The isotropic and anisotropic scattering difference curves were obtained $\Delta S(Q, t) = \Delta S_0(Q, t) - P_2(\cos \theta_Q)\Delta S_2(Q, t)$, where $\Delta S_0(Q, t)$ and $\Delta S_2(Q, t)$ were extracted via Legendre decomposition of the signal, considering the radial detector angle. The detected scattering signal was in the range $0.36 < Q < 4.34$ Å$^{-1}$. To obtain pair density information, the isotropic component, $\Delta S0(Q, t)$, was rebinned at $\Delta Q = 0.1208$ Å$^{-1}$, then multiplied by a Lorch window function, and inverted using the Natural Scattering Kernel (NSK) approach[35] with $\ell_2$ norm regularization to account for the inversion distortions originating from a finite detector range and scattering geometry. The anisotropic component, $\Delta S_2(Q, t)$, was not used for real-space inversion due to limited signal fidelity. Additional details on data reduction and analysis can be found in Supplementary Note 1.

### Excited state molecular dynamics of $Fe(CO)_5$

The methods for creating the trajectories used in this study were detailed in ref. 16. Briefly, The ab initio excited-state molecular dynamics (ESMD) simulations with surface-hopping were conducted using SHARC version 2.1[42] interfaced with ORCA quantum chemical package. The ESMD was carried out in the TDDFT framework accounting for 10 singlet states employing CAM-B3LYP functional and def2-TZVP basis set along with RIJCOSX approximation to make the computation faster. The approach generated an absorption spectrum from 7 singlet states through Wigner distribution of 300 phase space points based on ground state vibrational normal modes. This approach prioritized the $S_6$ state, aligned with the energy range of the typical UV pulses used in experiments, by limiting the inclusion of higher excited states to prevent deviation to lower wavelength regions. The study in ref. 16 utilized 116 out of 300 Wigner sampled[43] trajectories that successfully transitioned to the $S_6$ state (indicative of MLCT characteristics). These trajectories, extended to consider 10 singlet states and did not consider triplet states, consistent with the singlet-exclusive pathway for gas phase photodissociation of $Fe(CO)_5$ found in experiment[12]. Trajectory simulations ran up to 640 fs but were often terminated earlier (average termination time: 400 fs) due to SCF or gradient convergence failures, typically soon after Fe-CO bond dissociation. Non-adiabatic couplings for ESMD were accounted for via a local diabatization approach based on wavefunction overlap[44,45]. A 0.5 fs time step was used, along with energy-difference-based corrections[46,47] for electronic decoherence. Velocity rescaling was employed to adjust kinetic energy, while surface hops utilized SHARC's standard hopping probabilities. No additional scaling factors or damping terms were applied, and all other SHARC parameters were set to default values. Of the 116 trajectories, 103 met criteria allowing extrapolation to 640 fs using a dissociative atomic rotational-translational model (Supplementary Note 2, Supplementary Figs. 3–5). We required termination times of at least 210 fs and a minimum of 120 fs between dissociation onset and termination. This enabled us to extrapolate the dissociation dynamics of prematurely terminated trajectories, average the total simulated PD dynamics, and compare the results with experimentally measured PD dynamics at CO dissociation distances in the range $\gamma$.

## Data availability

The data generated in this study have been deposited in a git repository accessible at[48] and are provided as Supplementary Data.

## Code availability

The codes developed in this study have been deposited in a git repository accessible at[48].

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

## Acknowledgements

This work was supported by the US Department of Energy, Office of Science, Office of Basic Energy Sciences, Chemical Sciences, Geosciences and Biosciences Division. A.S., P.H.B, K.J.G, R.H. K.L. A.K. K.K. J.T.O. R.M.P. A.L.W. M.R.W. T.J.A.W and A.N. were supported by the U.S. Department of Energy, Office of Science, Basic Energy Sciences (BES), Chemical Sciences, Geosciences, and Biosciences Division, AMOS Program. E.B. work was supported by the U.S. Department of Energy, Office of Science, Basic Energy Sciences, Chemical Sciences, Geosciences, and Biosciences Division, Condensed Phase and Interfacial Molecular Science program, FWP 16248. P.M.W. acknowledges funding by the U.S. Department of Energy, Office of Science, Basic Energy Sciences, under award number DE-SC0017995. M.O. acknowledges financial support from the Swedish Research Council (VR contract 2021-04521). The authors gratefully acknowledge support from the Linac Coherent Light Source, SLAC National Accelerator Laboratory, which is supported by the US Department of Energy, Office of Science, Office of Basic Energy Sciences, under contract no. DE-AC02-76SF00515. We acknowledge the help of A.L. Wang-Holtzen and M.R. Ware with the experimental beam-time support.

## Author contributions

E.B., S.B., P.H.B., S.C., K.J.G., J.G., R.H., K.L., A.K., J.E.K., K.K., T.J.L., M.L., M.P.M., J.T.O., R.M.P., F.P., S.N., P.M.W., T.J.A.W., A.N. prepared and conducted the experiment at the Linac Coherent Light Source. A.S., E.B., A.N. analysed the experimental data. J.M.R., B.S., M.P.M., P.M.W. designed the scattering cell, R.M.P., A.B,. M.O. A.N. performed simulations, A.B, M.O. performed ab initio calculations. A.N. analyzed and interpreted simulation results and wrote the article with contributions from all authors.

## Competing interests

The authors declare no competing interests.
