## [Transparent Peer Review file · Nature Communications]

Real-space observation of the dissociation of a transition metal complex and its concurrent energy redistribution

Corresponding Author: Dr Adi Natan

Version 0:

Reviewer comments:

Reviewer #1

(Remarks to the Author)

Schori et al. employ ultrafast x-ray scattering (UXS) to follow the photoinduced nuclear dynamics of $\text{Fe}(\text{CO})_5$. The data is interpreted in the context of semi-classical excited state molecular dynamics. Together, the experiments and simulations reveal a detailed picture of the nuclear and electronic events following light excitation, and to some extent also quantify the distribution of the photon's energy into different (translational, vibrational, and rotational) modes.

There is excellent synergy between the computational and experimental work. The topic and results of this manuscript are likely to be of broad appeal to the readership of Nature Communications. The readability of the manuscript has improved considerably after recent changes that the authors made, such as including Fig. 1b and some explanatory text.

I recommend publication of this manuscript essentially as-is.

(Please note the minor typo appearing near the beginning of the introduction, "implications521")

Reviewer #2

(Remarks to the Author)

The authors have addressed all my comments of the previous review with care. I believe that the paper is appropriate for the audience of this journal and recommend publication as is.
